# The Cross Talk between Cellular Senescence and Melanoma: From Molecular Pathogenesis to Target Therapies

**DOI:** 10.3390/cancers15092640

**Published:** 2023-05-06

**Authors:** Jiahua Liu, Runzi Zheng, Yanghuan Zhang, Shuting Jia, Yonghan He, Jing Liu

**Affiliations:** 1Laboratory of Molecular Genetics of Aging and Tumor, Medical School, Kunming University of Science and Technology, Kunming 650500, China; 2Key Laboratory of Healthy Aging Research of Yunnan Province, Kunming Institute of Zoology, Chinese Academy of Sciences, Kunming 650201, China

**Keywords:** melanoma, skin aging, senescence, senolytic

## Abstract

**Simple Summary:**

Melanoma is a dreadful type of skin cancer with a poor response to therapy. Senescent cells have been found to accumulate with age and may contribute to the occurrence and development of melanoma. This review aims to provide an overview of skin aging and melanoma, including the molecular mechanisms, microenvironmental factors, and therapeutic strategies. In addition, some potential common targets between melanoma and cell senescence will be also discussed.

**Abstract:**

Melanoma is a malignant skin tumor that originates from melanocytes. The pathogenesis of melanoma involves a complex interaction that occurs between environmental factors, ultraviolet (UV)-light damage, and genetic alterations. UV light is the primary driver of the skin aging process and development of melanoma, which can induce reactive oxygen species (ROS) production and the presence of DNA damage in the cells, and results in cell senescence. As cellular senescence plays an important role in the relationship that exists between the skin aging process and the development of melanoma, the present study provides insight into the literature concerning the topic at present and discusses the relationship between skin aging and melanoma, including the mechanisms of cellular senescence that drive melanoma progression, the microenvironment in relation to skin aging and melanoma factors, and the therapeutics concerning melanoma. This review focuses on defining the role of cellular senescence in the process of melanoma carcinogenesis and discusses the targeting of senescent cells through therapeutic approaches, highlighting the areas that require more extensive research in the field.

## 1. Introduction

Melanoma is a malignant tumor that mainly occurs on the skin and originates from melanocytes. Although melanoma accounts for only a fraction of skin cancer-related cases, it accounts for over 80% of skin cancer-related deaths and is acknowledged in the literature as the most dangerous form of skin cancer [1]. The incidence of melanoma varies in many aspects, including region, skin color, gender, and age. The incidence of melanoma is much higher for older people than that for young people [2] (Table 1). As individuals ages, so too do their organs. The skin is the organ that is most susceptible to the aging process, as a result of numerous external factors. The aging of the skin leads to the reduction in the body’s immune function and increases the risk of age-related diseases, including melanoma [3]. Numerous associations between the development of melanoma and the skin aging process exist. Age and ultraviolet-light exposure are both important factors that cause skin aging and trigger the development of melanoma [4]. However, at present, little is known in the field about how skin aging affects the occurrence, development, and treatment of melanoma.

Senescent cells are a prominent feature of the skin aging process, where skin aging promotes their generation. Several types of senescent cells accumulate in the skin as the individual ages [5]. Melanocytes are present in the epidermis layer and contribute to the body’s protection against the damaged caused by ultraviolet (UV) radiation. However, with the increase in age, melanocytes undergo the stages of senescence or dysfunction, which causes the body to be more susceptible to DNA damage caused by UV radiation or oxidative stress, which are also major causes of the occurrence of cellular senescence [6].

The selective elimination of senescent cells (senolytics) and the modulation of proinflammatory senescence-associated secretory phenotypes (senostatics) have been shown, in the literature to alleviate age-related diseases in various animal models and improve cancer outcomes [7,8]. In a mouse melanoma model, adjuvant treatment performed with senolytics significantly reduced the tumor’s volume and prolonged the survival outcome [9]. Although senolytics clearly perform well as a form of adjuvant therapy for tumors, limited research on the relationship between senolytics and melanoma exists in the literature, and further clarification is required regarding the prevalence, selectivity, resistance, and toxicity of senolytics in a preclinical setting [10]. Therefore, the exploration of the pathogenesis of melanoma and the role senescence plays in this context is important for improving the selectivity of targeting harmful senescent cells, enriching potential targets, and reducing resistance. For this purpose, we discuss the role of cellular senescence in the formation of melanoma, the process of metastasis, and treatment outcomes. In addition, the potential common targets that exist between them are also discussed (Figure 1).

## 2. Characteristics of Cellular Senescence

Senescence describes a particular cell phase, which was previously considered in the literature to be a permanent state of cell cycle arrest. However, recent studies have shown that senescent tumor cells can escape senescence under certain conditions and undergo persistent rather than permanent cell cycle arrest. It may be related to the inhibition of hTERT expression by overexpression of oncogenes, loss of p53/p16INK4a, and change of microphthalmia-associated transcription factor (MITF) expression. In a hypothetical model proposed by Tareq Saleh et al., therapy-induced senescent (TIS) tumor cells avoided senescence and recovered the ability to self-renew following a period of growth arrest and autophagy-mediated homeostasis [11].

The molecular characteristics of cellular senescence mainly include DNA damage and telomere shortening, epigenetic changes, proteostasis disruption, and senescence-associated secretory phenotypes (SASPs). SASP refers to a group of factors secreted by senescent cells, including inflammatory factors, growth factors, chemokines, and matrix metalloproteinases, which can affect the function and structure of the surrounding cells and tissues. SASP can enhance or propagate the stage of cellular senescence through autocrine or paracrine signaling, and is considered in the research to be the main factor that causes chronic inflammation to occur and promotes tumor progression [12]. Senescent melanocytes are the source of melanoma, and, in addition to other aging cells present in the skin (such as dermal fibroblasts and immune cells), also play an important role in the occurrence, development, and treatment of melanoma [13]. Therefore, we further discussed the role of different types of senescent cells in relation to the progression of melanoma.

## 3. Developmental Stages and Genetic Characteristics of Melanoma

The formation of melanoma is the result of complex interactions occurring between exogenous and endogenous triggers as well as tumor-intrinsic and immune-related factors [14]. Melanoma has the highest mutation rate among all forms of cancer, which may be caused by the exposure to ultraviolet radiation, as numerous UV signature mutations can be observed in the restricted mutation sites of melanomas. Therefore, melanoma has numerous subtypes of tumor cells present within the same melanoma, making it a highly heterogeneous tumor [15,16].

Due to the high heterogeneity level of melanoma, a single treatment modality cannot remove all cancer cells from the body. Following the appropriate treatment and surviving it, these cancer cells become resistant to the initial treatment modality and form a major subpopulation of recurrent melanomas [17]. Melanoma is a form of skin cancer closely related to the genetic changes that occur in the human body. To develop more effective targeted drugs in the relevant field of study, we must understand the changes that occur in genomes and transcriptomes during the pathogenesis of melanomas. In most cases, the development of melanoma follows the following four stages, according to Dorothy C. Bennett’s model: the occurrence of mitogenic driver mutations (mutations that occur during cell division such as *BRAF* mutations) and nevus growth; avoiding of senescence and radial growth stages; overcoming apoptosis and vertical growth stages; and immortality and metastasis outcomes [18]. These processes involve the occurrence of numerous changes in the genomes and transcriptomes. The genomics and transcriptome analyses conducted by A. Hunter Shain et al. showed a sequence of changes occurring in primary melanoma, primarily mutations in the *BRAF*, *RAS*, or *NF1* genes upstream of the mitogen-activated protein kinases (MAPK) pathway. Secondly, telomerase upregulation causes cells to become immortal, and then chromatin landscape regulation allows for the cells to avoid immune surveillance. The G1/S checkpoint coverage process causes the cells to uncontrollably proliferate, MAPK signaling increases to create more invasive cells, and, finally, the P53 and PI3K pathways are destroyed in order to resist the process of apoptosis [19]. Among these processes, one of the most important signaling pathways is MAPK, which plays a key role in establishing and transforming malignant melanocytes. Large-scale genomic studies have identified four molecular subclasses of cutaneous melanomas: the *BRAF*, *RAS*, and *NF1* mutants, and triple WTs (wild-types). These subclasses display high MAPK activity levels. However, the activation mechanisms present were different. *BRAF* and *RAS* mutations cause these gene products to autonomously activate the MAPK pathway without the presence of upstream activators. Neurofibromin (NF1) inhibits RAS activity and NF1 inactivation promotes the development of melanoma. A triple WT melanoma exhibits high levels of ERK (a downstream effector molecule of the MAPK pathway)activity without the presence of all three mutations. Therefore, there may be other more appropriate methods to successfully activate the MAPK pathway [20].

Melanocytes with mitosis-driven mutations usually develop into benign tumors (such as moles or polyps) under the restriction of cellular senescence, whereas a melanoma is a malignant tumor that has avoided senescence. The main triggers of senescence are telomere shortening, oncogene overexpression, drug treatment, and stress-induced DNA damage signals. The p53 and p16 pathways are the main pathways that activate cellular senescence. Avoiding cellular senescence is an essential stage in the development of melanomas. In human melanoma cells, the inactivation of the p16 pathway (encoded by the *CDKN2A* gene) and mutations occurring in the telomerase reverse transcriptase (*TERT*) promoter may be required for the melanoma to successfully avoid senescence [21].

*CDKN2A* is a gene that presents the highest genetic alteration rate in melanomas. It encodes two important tumor suppressors: p16 and ARF (p14). *CDKN2A* is often deleted, mutated, or silenced by methylation in most melanomas. The loss of tumor suppressors is a key factor in the progression of melanoma [22,23]. Telomerase activation is the main mechanism that allows the melanoma to maintain its telomere length. Mutations occurring in the *TERT* promoter are one of the earliest secondary changes that occur in approximately 75% of melanomas. However, TERT activation alone is not enough to maintain the telomere length in melanoma cells. Recent studies have determined that mutations that occur in the promoter of Tripeptidyl-peptidase 1 (TPP1), a telomere-binding protein, coexist with TERT mutations almost simultaneously. The two mutations synergistically promote telomere maintenance in melanomas [24]. In addition, some studies have observed that a small portion of melanomas use an alternative mechanism called the alternative lengthening of telomeres (ALTs) to extend their telomeres by a homologous recombination process. However, few studies on how ALT works in melanomas exist, to date [25,26]. In addition to the genes previously mentioned, other genes, such as *PTEN*, *PPP6C*, *CDKN2B*, *TBX2*, and *MITF*, also frequently mutate in melanomas and participate in their avoidance of a malignant transformation during senescence [18,27,28].

In addition to the genetic changes occurring in melanoma cells, other cells in the environment also contribute to the development of melanomas. Keratinocytes are the cells that are adjacent to melanocytes, and melanomas can induce the downregulation of the Desmoglein-1(Dsg1) protein in keratinocytes by the paracrine signaling process. Dsg1, a specific desmosomal cadherin expressed by keratinocytes, has the ability to bind to the Erbb2-interacting protein, thereby inhibiting the EGFR-MAPK signaling pathway [29]. The down-regulation of keratinocyte Dsg1 not only inhibits Braf-mutation-induced OIS, but also promotes melanoma cell migration activity [30,31]. Moreover, melanoma-derived exosomes induce normal fibroblasts (Nfs) to transform into cancer-associated fibroblasts (CAFs), which can adapt to different types of melanoma cells by altering their biology and behavior. CAFs can also secrete factors that stimulate tumor growth, angiogenesis, and drug-resistance activities, thereby supporting tumor cell proliferation, neovascularization, and resistance to therapy [32,33,34].

Melanomas not only avoid senescence, but also possess a strong antiapoptotic ability, which is another important feature of its development. In p16-deficient melanocytes, apoptosis is a key factor that limits their malignant transformation. These cells present a high level of apoptosis when cultured, accompanied by increased levels of p53 and p21. However, adjacent keratinocytes inhibit the death of p16-deficient melanocytes by secreting SCF (stem cell factor) and EDN1 (endothelin 1) [35]. Therefore, multiple antiapoptotic changes must occur if p16-deficient melanocytes are to successfully exit from the epidermis and deteriorate further. In fact, mutations or the abnormal expression of numerous antiapoptotic-related genes have been observed in the studies conducted on advanced melanomas, such as *APAF-1*, *PTEN*, *PREX2*, and *TP53* [18]. Apoptotic protease-activating factor 1 (APAF-1) is an important apoptotic effector located downstream of Rb, which is activated in the absence of p16. However, APAF-1 expression is significantly reduced in malignant melanomas. Increasing or restoring APAFP-1 expression in malignant melanomas may enhance the killing effect of chemotherapeutic drugs on tumor cells and overcome chemoresistance [36,37]. Phosphatidylinositol 3,4,5-trisphosphate 3-phosphatase and dual-specificity protein phosphatase (PTEN) not only helps melanoma cells bypass senescence, but also endows them with an antiapoptotic ability. PTEN is an inhibitor of AKT(protein kinase B), whose inactivation leads to the upregulation of AKT and the activation of related antiapoptotic and mTOR-signaling pathways. Approximately one-quarter of melanomas present PTEN loss or an abnormal expression. In addition, frequently activated mutations in the *PREX2* gene have been observed in melanomas, in the research, which further increase the possibility of PTEN loss occurring in melanomas [38]. Unlike other types of cancer, the *TP53* gene only expresses a 10% mutation rate, approximately, in melanomas. However, MDM4 protein upregulation activity was observed in approximately 65% of I–IV-stage human melanomas, which is one of the p53 antagonists [39]. This may explain why *Tp53* presents a low mutation rate in melanomas.

In addition, melanomas also have a strong immune suppression ability and increased resistance to ferroptosis. Melanoma cells can express some inhibitory molecules, such as Programmed cell death 1 ligand 1 (PD-L1) and Cytotoxic T-lymphocyte protein 4 (CTLA-4), to block T-cell activation and function activities [40,41]. For example, Secretogranin II (SCG2) was overexpressed in melanoma tissues obtained from patients with a poor prognosis, and promoted immune escape by downregulating the expression of MHC-I ((major histocompatibility complex I)) [42].

In the research, at present, targeted therapies utilized for melanomas mainly focus on the MAPK pathway, which is unfavorable for improving the response rates produced and reducing drug-resistance behavior. Depending on the significant changes evident during the development process, the development of additional targeted drugs, followed by the use of precision medicine to precisely target different types of melanomas, has important implications for treatment outcomes.

## 4. Mechanisms and Effects of Senescent Cells in the Development and Progression of Melanomas

The skin is a complex organ composed of various types of cells, mainly consisting of the epidermis, dermis, and hypodermis. With aging, the number of senescent cells present in the skin significantly increases, including melanocytes, fibroblasts, keratinocytes, and immune cells [5]. Senescent cells are characterized by the loss of their proliferative potential and the secretion of age-related secretory phenotypes (SASPs). In addition, similar activities are often evident, such as increased cell apoptosis and iron death resistance, increased lysosomal count and secretion, the epigenetic modification of chromatin, and immune escape [41,43,44,45].

Senescent cells present different double-sided effects on melanomas. On the one hand, the transient presence of senescent cells may be beneficial in the treatment of melanomas. Senescent cells are immunogenic and stimulate the clearance of immune-mediated tumor cells [46]. For example, CCL5 (Chemokine (C-C motif) ligand 5) secreted by Aurora kinase A (AURKA) or CDK4/6 inhibitor-induced melanoma senescent cells promotes the recruitment of tumor-infiltrating leukocytes (TILs), enhancing their killing effect on tumor cells [47]. Moreover, senescent cancer cells may be used in vaccines to prevent cancer occurrence. In mouse models, healthy mice were inoculated with senescent cancer cells after preventing or delaying tumor formations [48].

On the other hand, the continuous accumulation of senescent cells can promote the occurrence and development of melanomas and affect treatment, and SASP plays a key role in this process [49]. In addition, studies have shown that SASP inhibits the scavenging effect of macrophages on dermal senescent fibroblasts, further promoting the accumulation of senescent cells [50]. PD-L1 has been observed to be heterogeneously expressed in some senescent cells. These senescent cells are not only resistant to T cells but also possess a stronger SASP phenotype [45]. Different senescent cells play different roles in the process of melanoma development. Therapy-induced senescent (TIS) tumor cells are likely to be the source of cancer recurrence and participate in tumor metastasis and drug-resistance processes. In the secretomes of *MITF*-silenced senescent melanoma cells, E-cadherin was reduced and CCL2 (Chemokine (C-C motif) ligand 2) was significantly increased. CCL2 stimulated the invasion ability of naïve melanoma cell in organs (not treated with any drugs or exposed to therapy), and both occurrences jointly promoted melanoma tumor formation and metastasis activities [51]. The supernatant of A375 melanoma senescent cells induced by cisplatin in vitro activated the ERK1/2-RSK1 pathway to promote the growth of normal nonsenescent A375 cells [52]. Increased studies have observed that secretomes obtained from TIS senescent melanoma cells activated the STAT3 pathway, causing melanoma cells to transition into a mesenchymal phenotype, which facilitates chemoresistance and recurrence behaviors [53].

In addition to tumor cells, senescent cells increase significantly in skin with age, including melanocytes, fibroblasts, keratinocytes, and immune cells. In the research, they have also been shown to be involved in melanoma occurrence and development processes. In middle-aged individuals, senescent melanocytes (SMCs) account for 5% of all melanocytes present in the body, and SMCs in exposed skin increase significantly in individuals older than 61 years [5]. SMCs produce increased DNA damage signals and senescence in the surrounding cells by paracrine secretion behavior. This process may be caused by the IP-10/CXCR3/mitochondrial ROS pathway. By clearing senescent melanocytes, ABT-737 (a type of senolytics) or mitochondrial-targeted antioxidant MitoQ prevent the occurrence of epidermal atrophy induced by senescent melanocytes and restore keratinocyte numbers [6].

As previously mentioned, keratinocytes can help melanoma cells resist apoptosis and aging processes, and senescent keratinocytes (SKCs) promote tumor migration activity in cutaneous squamous cell carcinomas (cSCCs). However, the role of SKCs in the occurrence and development of melanomas is rarely studied in the literature. Studies to date suggest that they may also be involved in the invasion of melanoma cells [54]. Senescent human dermal fibroblasts (SHDFs) promote the development and progression of melanomas [55]. Skin fibroblasts are the main cells present in the dermis. Normal HDFs secrete many components involved in the formation of the extracellular matrix (ECM) and regulate the survival and proliferation behavior of naïve T cells (not experiencing secondary lymphoid organ activation and differentiation) in the skin by releasing signal molecules, such as IL-7 and CCL19 [56]. Senescent fibroblasts not only reduce the ECM-related gene expression and increase MMP expression, but also present adipogenetic characteristics and reflect these changes through secretomes [57]. HAPLN1 (a protein that links hyaluronic acid and proteoglycan) is one of the most downregulated proteins present in aging fibroblasts. HAPLN1′s deletion of ECM not only increases the invasion ability of melanoma cells but also prevents the occurrence of T-cell infiltration into tumor sites. The restoration of HAPLN1 can improve mononuclear immune cell mobility and cause melanomas to metastasize to lymph nodes more than to visceral organs. Lymph node metastasis is easier to cure by performing surgery or other methods than visceral metastasis [58,59]. In addition, aging dermal fibroblasts also secrete sFRP2 (a Wnt antagonist) and neutral lipid secretion, which enhance melanoma drug-resistance effects (see below) [60,61].

Senescent cells have promoted melanoma development and this mechanism is related to the aging of the immune system. Senescent cells can induce the aging of the immune system by secreting proinflammatory cytokines and chemokines. The aging of the immune system affects the body’s ability to resist tumors, leading to the occurrence, development, and metastasis of tumors. In other studies, there was a reduction in the number, function, and diversity of T cells; the expression of MHC-1 (Major histocompatibility complex I); and the activity of natural killer cells, macrophages, and dendritic cells, which affected the tumor immune surveillance outcome. In addition, the aging of the immune system results in a weakened immune system, thus reducing the body’s ability to resist tumors [62,63]. In addition, senescent stromal cells reshape the tumor microenvironment by secreting related factors through SASPs, forming local immune suppression regions, and recruiting myeloid-derived suppressor cells (MDSCs) (immune suppressor cells that can inhibit CD8 +T-cell functions), such as increasing the number of M2 polarized macrophages, inhibiting immune clearance, and promoting cancer progression activities [64]. In summary, delaying the cell-aging process, reducing SASP secretion, and targeting senescent cells, such as SHDF, SMC, and TIS melanoma cells, may help inhibit the progression rate of melanomas.

## 5. The Role of Senescent Cells in Melanoma Therapy

At present, therapy used for treating melanomas faces numerous challenges, such as response-rate issues, antiresistance factors, and an attempt at reducing toxicity. Senescent cells play an important role in melanoma treatment, affecting both tumor development and treatment outcome and prognosis factors. First, senescent cells may be the source of cancer recurrence and resistance [65]. In a study, Xue et al. divided the formation of melanoma-resistant behavior into three stages: early survival, reversal of aging, and irreversible resistance. They observed that differences in the levels of MITF and Interferon-gamma (IFN-γ) led to innate resistant melanoma cell subpopulations MITF (Extremelyhi) and MITF (low) cell subsets have two types of melanoma cells that present different characteristics and responses to therapy. MITF (Extremelyhi) cells are resistant to MAPK inhibitors, and MITF (low) cells are poorly differentiated and antiapoptotic, which means that they have lost their normal functions and can resist cell death. They can also survive in harsh conditions, such as harsh treatment and nutritional stress, which makes it difficult to eliminate them. Similarly, the IFN-γ pathway has a similar effect. The excessive activation of IFNγ leads to the upregulation of immune checkpoints and dedifferentiation of melanoma cells. The expression of IFNγ in these two tumors leads to melanomas becoming resistant to immunotherapy [66]. These cells enter a preprogrammed aging state when receiving immune- or targeted therapy, which then promotes the tumor microenvironment through SASPs. SASP factors not only stimulate their own growth, invasion, and transformation processes and that of neighboring melanoma cells, but also induce immune suppressive cells (such as M2 polarized macrophages), thereby inhibiting immune clearance outcomes. Over time, these TIS melanoma cells reverse the aging process under the influence of epigenetic changes and accumulate mutation genes (such as activating mutations of *NRAS* and altered the expression of *TP53* family isoforms) [67], and highly expressed multiple stemness-related genes, eventually leading to tumor recurrence and resistance behaviors [17,51,68]. Therefore, the elimination of TIS tumor cells has a positive effect on melanoma treatment outcomes.

In addition to senescent cancer cells present in the tumor itself, the senescence of matrix, immune, and other cells contributes to melanoma-resistance behavior. Metabolic reprogramming refers to the change in energy metabolism levels (glycolysis, protein metabolism, lipid metabolism, etc.) by tumor cells, in order to adapt to the external environment. It is an important mechanism for melanomas to resist attacks on the immune system and drug treatments. In a study conducted by Salzer MC et al., it can be observed that lipid production, lipid metabolism, and adipocyte differentiation-related genes were significantly upregulated in aging skin fibroblasts [57]. Alicea Gmet al. observed that aging fibroblasts increase the production of neutral fatty acids in the body, which are absorbed and converted into energy by melanoma cells through the fatty acid transporter (FATP2), protecting cancer cells themselves from BRAF-targeting drugs [61]. In addition, it was also observed that senescent fibroblasts secreted a Wnt signaling pathway antagonist sFRP2, which activated NF-κB and β-catenin signaling pathways and reduced the DNA damage repair enzyme APE1 level. APE1 deficiency weakened the melanoma cell response to DNA damage caused by reactive oxygen species, thereby increasing resistance to targeted therapy [60]. Moreover, a high number of senescent immune T cells were also observed in melanomas [69]. T-cell functional status determines antitumor immunity and immunotherapy effects. Senescent T cells not only failed to effectively recognize and respond to tumor antigens, but also released factors that promoted or inhibited immune responses (such as IL-2, IL-6, IL-8, TNF, IFN-γ, IL-10, and TGF-β), thus enhancing the immune suppression behavior present in the tumor microenvironment [70,71].

As previously mentioned, the application of senolytics or senostatics as adjuvants to other treatments during melanoma therapy, selectively eliminating harmful senescent cells and SASPs at appropriate times, has positive implications for melanoma treatment outcomes. Several studies have verified the effectiveness of this strategy in mouse melanoma models. The senolytics cocktail Dasatinib + Quercetin (DQ) is the first effective senolytic product that can synergize with radiotherapy (RT) to eliminate senescent melanoma cells, reduce melanoma volume, and prolong mouse survival rates, enhancing the antitumor effect of RT in a time-dependent manner [72]. In addition, ES2 (a peptide that can interfere with the FOXO4-P53 signaling pathway and induce apoptosis) is a senolytic product that specifically eliminates senescent cancer cells, delays tumor growth, and improves survival outcomes when combined with prosenescence agents [73]. Moreover, other studies have shown that applying senostatics (resveratrol) can reduce the amount of SASP factors released by senescent cells and inhibit the stimulatory effect of senescent fibroblasts on melanoma cell proliferation and invasion, as well as reduce the expression of epithelial–mesenchymal transition markers associated with malignant characteristics [74]. Furthermore, whether or not the selective elimination of other senescent cells is beneficial for melanoma treatment has not yet been studied in the literature (Table 2).

## 6. Perspectives

The occurrence and development of melanomas involve a variety of genetic changes, some of which are strongly correlated in the formation of melanomas. Focusing on the specific mechanisms of these genes in the development of melanomas and the drugs targeting these genes or related pathways is of great significance for increasing drug-selection options and improving patient response rates. Numerous studies have shown that senescent cells promote melanoma progression rates and induce a resistance to drugs and poor prognosis outcomes. Although senolytics have great potential value as adjuvant cancer therapies, different types of senescent cells may produce opposite effects during melanoma treatment. In addition, the senolytic activity of senolytics on senescent tumor cells is highly dependent on the cell line and mode of senescence induction (senescence inducers). Therefore, clarifying the role of different senescent cells in the presence of melanomas and improving the selectivity for harmful senescent cells are issues that need to be addressed in the literature. At present, few studies concern the subject of the use of senolytics for the treatment of melanomas; therefore, whether eliminating senescent cells is ultimately beneficial for melanoma treatment and whether it will result in additional toxic side effects requires further research. At present, the main mechanisms of action concerning senolytics focus on antiapoptotic pathway (SCAP)-related proteins, such as P53, Bcl-2/Bcl-xL, and heat-shock protein 90 (Hsp90), which are more susceptible to resistance due to overconcentrated targets. Therefore, we believe that, in the future research conducted on senolytics, improving the selectivity for harmful senescent cells, diversifying targets, and improving drug delivery methods are some directions that must be considered. Melanomas and cellular senescence share numerous similarities. For this reason, we summarized some potential common targets they share.

## Figures and Tables

**Figure 1 cancers-15-02640-f001:**
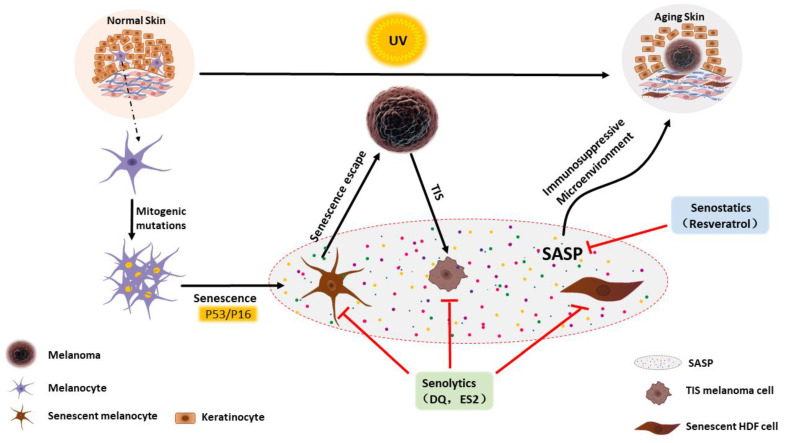
Melanocytes and keratinocytes in the skin can provide protection against oxidative damage caused by UV exposure. However, excessive UV exposure can not only damage skin cells and promote skin aging but also increases the probability of the occurrence of melanocyte mutations, thereby increasing the risk of melanoma. The process of melanocytes developing into melanomas usually includes the following stages: malignant proliferation caused by a mitotic mutation, cellular senescence, and senescence escape. In the process of skin aging, senescent cells, such as melanocytes and fibroblasts, also gradually accumulate. In addition, some treatment methods may also induce the senescence of melanoma cells. These senescent cells create an immunosuppressive microenvironment by senescence associated secretory phenotype (SASPs), which promote the occurrence, development, and drug-resistance behaviors of melanoma cells. Therefore, drugs that selectively eliminate senescent cells (senolytics) or regulate SASP (senostatics) present potential effects for the treatment of melanoma. Some senolytic and senostatic drugs, such as DQ (Dasatinib + Quercetin), ES2, and resveratrol had a positive effect on the treatment of melanomas in different models. TIS: therapy-induced senescence.

**Table 1 cancers-15-02640-t001:** Incidence and mortality rates of cutaneous melanoma.

Continent	Males				Females			
	Incidence		Mortality		Incidence		Mortality	
	Age (Years) (0–59)	Age (Years) 60+	Age (Years) (0–59)	Age (Years) 60+	Age (Years) (0–59)	Age (Years) 60+	Age (Years) (0–59)	Age (Years) 60+
Oceania	14.6	202.9	0.88	24	14.1	116.4	0.58	10.8
North America	7.6	106.5	0.47	9.8	9.8	50.6	0.36	4
Europe	6.6	55.1	0.82	11.7	8.6	33.9	0.62	6.3
Asia	0.2	2.5	0.09	1.4	0.2	1.9	0.08	1
Latin America	1.2	13.4	0.33	5.2	1.1	10.6	0.22	3
Africa	0.29	5.6	0.09	2.6	0.36	5.5	0.1	2.5
World	1.4	23.1	0.22	4.6	1.7	13.7	0.17	2.7
	ASR(World) per 100,000							

Estimated age-standardized incidence and mortality rates for cutaneous melanoma occurrence. Data source: Globocan 2020; graph production: Global Cancer Observatory (http://gco.iarc.fr accessed on 8 February 2023).

**Table 2 cancers-15-02640-t002:** Common targets related to melanomas and cellular senescence.

Gene Symbol	Function		Reference
	Melanoma	Senescence	
*SRC*	Promotes melanoma cell growth and invasion.	Inhibits cell apoptosis and promotes cell senescence.	[75,76,77]
*EGFR*	Promotes melanoma cell invasion and drug resistance.	Promotes the acquisition of drug resistance phenotype and induces up-regulation of PD-L1 expression.	[78,79,80,81]
*AKT1*	Promotes melanoma metastasis and drug resistance.	Involvement in drug-induced cellular senescence.	[82,83,84,85]
*STAT3*	Participates in tumor growth, angiogenesis, EMT, metastasis, and tumor immunosuppression.	Promotes the production of CAF phenotype of senescent fibroblasts and resistances to cell senescence.	[86,87,88,89,90,91]
*TNF*	Promotes melanoma cell senescence, invasion, and immunosuppressive factor expression.	Causes cellular senescence.	[92,93,94,95,96]
*EP300*	Involves in the resistance of melanoma cells to BRAFi.	Drives cellular senescence phenotypes.	[97,98]
*IL6*	Promotes the growth, migration and resist apoptosis of tumor cells.	The important cytokine of SASP.	[99,100,101,102]
*HSP90AA1*	Promotes melanoma growth, survival, and immunosuppressive microenvironment.	Promotes cellular senescence and mitochondrial dysfunction.	[103,104,105,106,107,108]
*PIK3R1*	Melanoma proliferation, migration, invasion, and drug resistance.	Causes T-cell senescence.	[109,110,111,112]
*CD274*	Inhibits immune function.	Immune escape and SASP secretory.	[45,113,114,115,116]
*PTEN*	Melanoma invasion, metastasis, tumor cell escape and tolerance.	Accelerates cellular senescence.	[27,117,118,119,120]

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
