# Peer review of "The Cross Talk between Cellular Senescence and Melanoma: From Molecular Pathogenesis to Target Therapies"

_cancers, 2023, doi:10.3390/cancers15092640_

Round 1
Reviewer 1 Report
The review by Liu et al aims to summarize the role of cellular senescence in melanoma development, progression, and therapy but I found a lot of claims lack detailed elaboration and explanation. The information provided is rather fragmented and should be better organized and logically presented. The following points should be addressed to improve the quality of the review:
Should have a separate section to define the cellular and molecular characteristics of senescence before further elaborating on how it affects melanoma development, progression, and therapy.
There are many sentences that need further elaboration and explanation:
1. It was mentioned that senescent tumor cells can escape the senescence state under certain conditions and undergo persistent rather than permanent cell cycle arrest. Elaborate further under what conditions tumor cells can escape the senescence state and the differences between persistent and permanent cell cycle arrest.
2. The three sentences from paragraphs 46 to 49 have similar meanings. Combine them into one sentence.
3. Provide the full name of the continents in the legend of Table 1.
4. what do you mean by “mitosis-driven mutations”? Raise an example of how melanocytes develop into benign tumors by mitosis-driven mutations.
5. What is dsg1 protein? What paracrine factors cause the downregulation of Dsg1 protein expression in keratinocytes?
6. I have no way to understand the meaning of Figs 1 and 2. They are not self-explanatory just by looking at the diagrams. There are so many arrows that I don’t know what they mean as no explanation is provided in the legends.
7. The term “Melanoma senescent cells’ described in the text is contradictory to what was mentioned melanoma is a malignant tumor that has escaped from senescence.
8. Provide more information about CCL5 and CCL2
9. HAPLN1 downregulation in aging fibroblasts limits the ability of melanoma to metastasize, implying senescent fibroblasts are beneficial. However, it is also mentioned aging dermal fibroblasts secrete sFRP2 and neutral lipids to enhance melanoma drug resistance. Clarify whether aging/senescent fibroblasts are good or bad.
10. How do senescent cells induce immune system aging? What are the underlying mechanisms?
11. How do differences in MITF and IFN-gamma expression levels lead to innate resistance melanoma subpopulations?
12. Raise some examples to elaborate on how epigenetic changes and accumulated mutated genes could lead to tumor recurrence and resistance
13. Turnitin's report showed there is a 16 % similarity of the review content to the public sources that require attention.
14. The whole review requires English editing and proofing.
Reviewer 2 Report
This manuscript is well-written but the reviewer has some concerns.
Major points
1. The authors define that melanoma is a malignant tumor that has escaped from senescence (line 104). However, after line 191, the authors use “senescent melanoma cells” many times, which is confusing. Do these “senescent melanoma cells” derives from treatment-induced senescence? The authors had better explain in detail how, when and where melanoma cells or tumor cells become senescent.
2. The two figures are hard to understand: Cells and tumor tissues are mixed. Cell shape is not consistent by cell type. There are different meanings of arrows. What do the red and yellow dots in the stroma mean? The authors should revise the figures radically.
Minor points
1. Lines 64-65: grammatically incorrect.
2. Gene symbols had better be written in Italics.
3. Line 103: melanocytes do not develop cysts.
4. Lines 151-152: AKT?
5. Line 171: Skin is an organ.
6. Figure 1: What is UA?
7. “Senescent cells” are used many times but they are ambiguous and confusing (senescent melanoma cells, senescent melanocytes, senescent keratinocytes, etc). Every time they need to be clarified, especially after line 191.
8. Line 209: What is “naïve melanoma cell”?
9. Line 233: What is “natural T cell”?
10. Line 248: What is “MCH-1”?
11. Lines 252, 276: “Matrix cells” had better be “stromal cells”.
Round 2
Reviewer 1 Report
The authors have faithfully addressed all my concerns.